# Microwave Absorption Ability of Steel Slag and Road Performance of Asphalt Mixtures Incorporating Steel Slag

**DOI:** 10.3390/ma13030663

**Published:** 2020-02-02

**Authors:** Baowen Lou, Zhuangzhuang Liu, Aimin Sha, Meng Jia, Yupeng Li

**Affiliations:** 1School of Material Science and Engineering, Chang’an University, Xi’an 710064, China; loubaowen@chd.edu.cn; 2School of Highway, Chang’an University, Xi’an 710064, China; zzliu@chd.edu.cn (Z.L.); mengjia@chd.edu.cn (M.J.); liyupeng@chd.edu.cn (Y.L.); 3Key Laboratory for Special Area Highway Engineering of Ministry of Education, Chang’an University, Xi’an 710064, China

**Keywords:** steel slag, asphalt mixture, microwave absorption, road performance

## Abstract

Excessive usage of non-renewable natural resources and massive construction wastes put pressure on the environment. Steel slags, the main waste material from the metal industry, are normally added in asphalt concrete to replace traditional aggregate. In addition, as a typical microwave absorber, steel slag has the potential to transfer microwave energy into heat, thus increasing the limited self-healing ability of asphalt mixture. This paper aims to investigate the microwave absorption potentials of steel slag and the effect of its addition on road performance. The magnetic parameters obtained from a microwave vector network analyzer were used to estimate the potential use of steel slag as microwave absorber to heal cracks. Meanwhile, the initial self-healing temperature was further discussed according to the frequency sweeping results. The obvious porous structure of steel slag observed using scanning electron microscopy (SEM) had important impacts on the road performance of asphalt mixtures. Steel slag presented a worse effect on low-temperature crack resistance and water stability, while high-temperature stability can be remarkably enhanced when the substitution of steel slag was 60% by volume with the particle size of 4.75–9.5 mm. Overall, the sustainability of asphalt mixtures incorporating steel slag can be promoted due to its excellent mechanical and microwave absorption properties.

## 1. Introduction

Asphalt mixtures have long been widely applied for pavement construction. However, as the most common deterioration, cracks inevitably form due to the hardening and brittleness of aged bitumen, vehicle loading and environmental factors [1]. Further, continued development of cracks will eventually cause the degradation of asphalt pavement. On the other hand, as a typical viscoelastic material, the bitumen can flow through open micro-cracks when the viscosity is low, which is similar to a capillary flow [2]. The viscosity of bitumen reduces with an increase in temperature, and its thermoplastic characteristic make it possible for asphalt mixtures to heal or seal the cracks [3]. Due to a nonlinear relationship between viscosity of bitumen and temperature, bitumen can be regarded as a Newtonian fluid when the temperature achieves a certain threshold in the range of 30–70 °C [4]. Thus, it can be inferred that intrinsic healing properties of bitumen are greatly influenced by the temperature and elevated temperature contributes to the flow of bitumen [5]. Although self-healing is the inherent property of asphalt binder and some micro-cracks can be healed during the rest period, it is quite limited especially under repeated loading.

In order to accelerate the self-healing process of asphalt mixtures, several technologies have been applied to prolong the pavement service life. To date, three self-healing technologies including microcapsule technology [6], induction heating technology [7] and microwave heating technology [8] are well discussed and applied. The performance of microwave heating on adhesion and moisture damage of asphalt mixtures has been investigated since 1988 [9]. The effects of the morphology of microwave absorbers [10,11], the thermal expansion of bitumen [12], the addition of metal wastes [13,14] and other factors on microwave self-healing properties have later been discussed. When comparing these self-healing methods, microwave self-healing technology has specific advantages such as shorter heating time, higher heating efficiency and greater economy. Thus, the microwave heating vehicles have been developed and with the movement of vehicles, the microwave energy produced by microwave generator can be transferred into heat [15]. Several researches have indicated that with the addition of microwave absorbers, such as steel slags, carbon nanotubes and graphene, the microwave crack-healing properties can be greatly improved. The mechanism is that under the action of microwave radiation, the magnetic loss originated from eddy currents, domain wall resonance and natural resonance can generate internal friction heat, so that bitumen can flow through and recover cracks [16].

According to the World Steel Association report [17], approximately a hundred million tons of steel slag are produced by the steel industry every year, which accounts for more than 10% of raw steel output around the world. This by-product has adverse effects on land occupation, water resources and environment management. As an ideal secondary resource, steel slag has been widely applied in civil construction, including pavement, concrete masonry and building materials in recent years. Previous experiments proved that steel slag asphalt pavement is feasible due to its higher stiffness, better abrasion resistance, lower fatigue, and lower permanent deformation properties [18,19]. Hence, the utilization of steel slag in asphalt mixtures can be conducive to natural resource savings and release of environmental pressure. Meanwhile, steel slag is rich in metallic oxides such as MgO, CaO, Fe_3_O_4_ [20], thus it has been regarded as a kind of magnetic material. Many researchers have made great efforts to incorporate steel slag (as a microwave absorbing material) into conventional asphalt pavement for the purpose of self-healing [14,15,21,22]. As a result of magnetic components like iron and iron oxide present in steel slags, the values of electromagnetic parameters of steel slags are higher than those of traditional aggregates [23], making the steel slag a potential magnetic loss microwave absorber. 

Previous work has already ensured the feasibility of the application of steel slag in asphalt materials, but the possibilities of replacing partial traditional limestone aggregates with steel slags for self-healing purposes needs to be further investigated. First, morphological features of steel slag were observed by scanning electron microscopy (SEM). Then, as a typical microwave absorber, electromagnetic parameters of the steel slag obtained from the microwave vector network analyzer were discussed to evaluate its microwave sensitivity. Initial self-healing temperature was calculated by the flow behavior index via dynamic shear rheometer (DSR) frequency sweep test results. Finally, the comprehensive feasibility of steel slag asphalt mixture was studied from the perspective of the microwave-absorbing mechanism and various road tests, and the optimal substitution content of steel slag can be acquired based on the results of road performance tests.

## 2. Materials and Methods

### 2.1. Raw Materials

The raw materials include bitumen, aggregates and fillers. The basic bitumen was provided by the Korean bitumen production company, with density 1.031 g/cm^3^, penetration 67 and softening point 47.8 °C according to the “Standard Test Methods of Bitumen and Bituminous Mixtures for Highway Engineering” (JTG E20–2011). The limestone aggregates and fillers were provided by Shaanxi, China. The steel slag aggregates were supplied by Baogang Group. AC-13 asphalt mixtures were used in this research, and the particle size distribution was shown in Table 1.

### 2.2. Sample Preparation 

The properties of aggregates were shown in Table 2 according to “Test Methods of Aggregate for Highway Engineering” (JTG E42–2005). The bitumen content of all the mixtures was 5.0% by mass of aggregates based on the Marshall test results. Due to the greater density and porous structure of steel slag, it was used as a substitution (0%, 20%, 40%, 60%, 80% and 100%) of limestone aggregate by volume with the particle size of 4.75–9.5 mm based on the results of mix design (as detailed in Table 3). 

### 2.3. Methods 

#### 2.3.1. Morphological Characterization of Steel Slag

Morphology of steel slags used in this study were observed by a scanning electron microscope (SEM, MERLIN Compact). Images were taken at a voltage of 5 kV with 5.00 KX and 10.00 KX magnification scale. In addition, map-scanning of an electron probe micro analyzer was applied to estimate the composition in steel slag at a voltage of 15 kV, where denser distributions of elements in steel slag indicates higher proportion [21]. Energy-dispersive X-ray spectroscopy (EDS) was adopted under 15 keV in this study to determine the elemental composition of materials [24].

#### 2.3.2. Microwave Absorption Properties

In order to evaluate the inherent microwave absorption characteristics of steel slags, the electromagnetic parameters were acquired through a microwave vector network analyzer (PNA-N5244A; E5071C, Xi’an, China) based on the coaxial transmission/reflection measurement. The frequency range was set as 2–18 GHz. The steel slag was firstly ground into powder, and then mixed with paraffin wax by the weight ratio of 1:1. Finally, cylindrical toroidal samples were processed with dimensions 3.04 mm inner diameter, 7.0 mm outer diameter and 2.0 mm height.

#### 2.3.3. Initial Self-Healing Temperature

During the healing process, the rearrangement of molecules in the bitumen may cause the viscoelastic recovery response or thixotropicity. Besides, the mobility potential of the molecules in the bitumen contributes to the capability of diffusion [4], and higher molecular mobility is directly related to relatively high temperature. In order to find out the initial self-healing temperature, a dynamic shear rheometer (DSR) was used to process frequency sweep analysis. According to ASTM D7175–08 [25], the frequency sweep range was 0.01–10 Hz and fixed test temperatures were from 34 to 70 °C (at 6 °C intervals). The complex viscosity was measured via DSR using 25-mm parallel plate arrangement having a 1-mm gap.

#### 2.3.4. Wheel Tracking Test

Specimens with different steel slag content were mixed and compacted by the plate compactor in the optimum asphalt–aggregate ratio according to the Marshall test. The asphalt specimens with dimensions 300×300×50 mm were produced with three specimens per parallel control test. The temperature at which these tests were conducted was 60 °C (0.5 °C accuracy) and values of dynamic stability (DS) were tested under 0.7 MPa (0.05 MPa accuracy). The slab of the compactor may pass 24 times to and fro, and 9 kN load pressures applied, during compacting according to the JTG E20–2011 [26].

#### 2.3.5. Low-Temperature Bending Test

In order to evaluate the low temperature crack resistance of asphalt mixture, the maximum bending strain ($\varepsilon_{B})$ at the bottom of the beam was calculated based using the method T 0715 of JTG E20–2011. The prepared beam (250 mm×30 mm×35 mm) was first soaked at −10 °C for 45 min, and then a load was applied onto the center of the beam at the speed of 50 mm/min. The bending deflection of the beam was recorded by the sensor during the test, and the value of maximum bending tensile strain ($\varepsilon_{B}$) was measured.

#### 2.3.6. Water Stability

The residual stability test and freeze–thaw splitting test were selected to estimate water stability of asphalt mixtures according to the JTG E20–2011. All the test specimens were formed in accordance with Marshall’s method. The residual stability was calculated by Equation (1).
(1)$$\mathrm{RMS}=MS_{1}/MS_{0}\times 100$$
where RMS represents residual Marshall stability (%), $MS_{0}$ and $MS_{1}$ represent the Marshall stability of the specimen immersed in 60 °C water for 30 min and 60 °C water for 48 h (kN), respectively.

The freeze–thaw splitting strength ratio (%) was calculated by Equations (2) and (3) according to the JTG E20–2011.
(2)$$R_{T}=2P/\pi Dt$$
(3)$$\mathrm{TSR}=R_{T2}/R_{T1}\times 100$$
where $R_{T}$ is the splitting strength of specimens (MPa), and P, t, D is the maximum load (N), the height immediately before test (mm) and diameter (mm) of the specimen, respectively. $R_{T1}$ and $R_{T2}$ is the splitting strength of specimens before and after the freezing and thawing process (MPa).

## 3. Results and Discussions

### 3.1. Morphological Characteristics of Steel Slag

It can be clearly observed from Figure 1 that the structure of steel slag is porous and this structure may affect the mechanical properties of asphalt mixtures due to its high binder consumption. Besides, microwaves could also be reflected inside the pores of steel slag multiple times, generating heat efficiently due to more reflection loss. Meanwhile, the distribution of elements in steel slag were also illustrated in Figure 1, and the main elements were Ca, Fe, Si and Mg. Table 4 presented the mass concentrations of corresponding elements acquired from SEM/EDS results (shown in Figure 2). The proportion of Ca was more than 60%, this higher value may due to the high fluxes dosage for minimizing the impurities during the conversion of molten iron to steel [27]. Nevertheless, relatively high metal element content such as 17.19% of Fe, 4.66% of Mg, made steel slag a better microwave-absorber to transfer heat efficiently under the microwave irradiation.

### 3.2. Microwave Absorption Properties of Steel Slag

Microwaves are electromagnetic waves with frequencies in the range of 100 MHz to 100 GHz, while the corresponding wavelength is from 1 m to 1 mm. The electromagnetic wave loss occurs when it passes through materials and causes molecular motion. Thus, ideal microwave absorbing materials can absorb or dissipate microwaves and convert the microwave energy into heat efficiently. As for magnetic materials, there are two main mechanisms of electromagnetic wave loss, including dielectric loss and magnetic loss. The dielectric loss occurs mainly due to the conductivity and polarization loss [28], while the magnetic loss often comes from relaxation process during magnetization, such as exchange resonance, natural resonance, hysteresis and eddy currents [29]. 

The microwave absorption properties of an absorptive material are investigated using the relative complex permittivity ($\varepsilon_{r})$ and permeability ($\mu_{r})$, which can be expressed as follows.
(4)$$\varepsilon_{r}=\varepsilon^{\prime }-j\varepsilon^{\prime \prime }$$
(5)$$\mu_{r}=\mu^{\prime }-j\mu^{\prime \prime }$$

As shown in Figure 3, the electric and magnetic storage ability of steel slag can be reflected via the real parts of permittivity ($\varepsilon^{\prime })$ and permeability ($\mu^{\prime }$) of the specimen, while the electric and magnetic dissipation can be observed by the imaginary parts of permittivity ($\varepsilon^{\prime \prime })$ and permeability ($\mu^{\prime \prime }$) [30]. Therefore, higher $\varepsilon^{\prime \prime }$ and $\mu^{\prime \prime }$ help to attenuate the electromagnetic wave rapidly, thus transferring more microwave energy into heat. In Figure 3b, the imaginary part of permittivity ($\varepsilon^{\prime \prime }$) shows an overall downward trend in the frequency of 2–12 GHz, and exhibited the highest $\varepsilon^{\prime \prime }$ value at around 16 GHz. The imaginary part of permeability ($\mu^{\prime \prime }$) fluctuated in general (shown in Figure 3d), and had shoulder peaks at around 2.45 GHz (microwave frequency commonly applied), 6.1 GHz and 17.3 GHz frequency positions. From the above analysis, as a typical magnetic aggregate, the steel slag has relatively higher values of dielectric and magnetic conductivity loss, which can produce heat rapidly to heal cracks under microwave irradiation.

### 3.3. Initial Self-Healing Temperature based on Flow Behavior Index

As a typical viscoelastic material, bitumen can flow through the cracks and fill them above a temperature threshold due to its flexible and ductile properties [4]. This threshold temperature for flow is defined as the temperature when the flow behavior of bitumen changes from non-Newtonian to Newtonian [31]. Once the bitumen behaves as a Newtonian fluid, a pressure difference will appear in the contact points and promotes capillary flow of the bitumen, contributing to the closure of cracks [2]. Based on the power law relationship (shown in Equation (6)), the Newtonian flow characteristics of bitumen can be investigated [32].
(6)$$\eta^{*}=m{\left|f\right|}^{n-1}$$
where *f* is the frequency and $\eta^{*}$ is the complex viscosity acquired from DSR test results. The m and n are the fitting parameters and flow behavior index is defined as the dimensionless parameter n. The specimen corresponds to a Newtonian fluid while n = 1, and reflects a higher degree of pseudoplastic properties of the fluid when n < 1. Besides, the transition from 0.9≤n<1 is considered as near-Newtonian behavior [33]. Therefore, the temperature corresponding to n = 0.95 was chosen as the initial self-healing temperature in this study.

The relationship between frequency and complex viscosity of basic bitumen under various temperatures was shown in Figure 4. The complex viscosity presented a dropping trend under lower temperatures, and the curves became smooth with increasing temperature. The value of complex viscosity became constant when the temperature increased beyond 58 °C, and plateaued at a horizontal line at higher temperature, which meant that as the frequency changes, the complex viscosity remained stable. 

Based on the Figure 4 results, the fitting results of flow behavior indexes at different temperatures are shown in Figure 5. The value of n extracted from Table 5 was used to characterize the flow behavior index of basic bitumen and calculate its initial self-healing temperature. As the temperature increased, the bitumen flow behavior index increased and tended towards the Newtonian fluid state. According to the fitting equations, the initial self-healing temperature of basic bitumen is 46.4 °C and is slightly lower than the softening point of 47.8 °C, showing a promising flow property. Hence, the asphalt used in this study can start to flow and obtain a better self-healing performance even when the temperature is slightly lower than the softening point.

### 3.4. Road Performance

#### 3.4.1. High Temperature Stability

Dynamic stability is used to represent the rutting depth of pavement, while higher values indicate better stability and resistance to deformation under high temperature. The experimental results observed in Figure 6 showed that the value of DS reached a peak as the steel slag content increased continuously. When the replacement content of steel slag was up to 60%, the DS reached the maximum value (5732 times/mm), 1.45 times higher than the mixtures without steel slags. This phenomenon may due to the larger density, higher strength of steel slag and its closer contact with aggregates, which could enhance the integrity of the structure to resist high-temperature deformation. However, excessive steel slag content will increase the air voids and form more absorbed binder due to its porous structure. As a result, less effective binder may deteriorate the adhesion between bitumen and aggregates, thus, reducing the strength and high-temperature stability of asphalt mixtures. It reveals that the addition of steel slag with 60% substitution content contributes to a better permanent deformation response for asphalt mixtures.

#### 3.4.2. Low-Temperature Crack Resistance

In order to evaluate the low-temperature crack resistance of asphalt mixtures, the maximum bending strain (*ε_B_*) was measured via low temperature bending testing. Low-temperature cracking of asphalt pavement is caused by the brittleness and hardness of asphalt binder at low temperature [34], and higher values of *ε_B_* indicates better flexibility to resist cracks. As shown in Figure 7, the value of $\varepsilon_{B}$ showed a linear downward trend as the content of steel slag increased. However, the maximum bending strain over 80% steel slag content was lower than 2500 *µ_g_* which is considered as the specification requirement in China. This is due to the porous structure of steel slag; the air void content increased with the increase of steel slag content. The increasing substitution content of steel slag would absorb a large amount of asphalt binder, which may deteriorate the adhesion between aggregates. As a result, it is more likely to form cracks under the action of temperature stress. To sum up, the addition of steel slag impaired the low-temperature crack resistance of asphalt mixtures.

#### 3.4.3. Water Stability

Test results observed from Figure 8 indicate that both the value of residual stability and freeze-thaw splitting strength decreased with the increase of steel slag content. From the perspective of freeze–thaw splitting strength test results, the TSR can still reach up to 82.5% with a 100% steel slag replacement ratio, which was over 80% of that requested in the Chinese specification. However, the residual stability dropped to 79.2% when the steel slag replacement ratio was over 80%, which failed to meet the 85% request in the Chinese specification. This may be attributed to the increasing air void content introduced by steel slags, so that moisture is more easily able to enter the structure of asphalt mixture. When this occurs, the asphalt binder may move or even spall from the surface of aggregates, which could deteriorate the water stability of asphalt mixtures. Moreover, when the substitution of steel slag was 60%, the values of TSR and residual stability decreased only 5.25% and 7.24% compared with the asphalt mixtures without steel slag, which is within an acceptable range.

Based on the above analysis, it can be concluded that a certain amount of steel slag can improve the high-temperature stability of asphalt mixtures. Nevertheless, steel slag has a negative effect on both low-temperature crack resistance and water stability properties mainly due to its porous structure. Thus, there exists an optimal content of steel slag in asphalt mixtures to acquire better road performance, and in this study, the optimal replacement ratio of steel slag by volume is 60% with the particle size of 4.75–9.5 mm.

## 4. Conclusions

The main goal of this current study was to estimate the microwave absorption ability of steel slag and the road performance of asphalt mixtures incorporating steel slag. Based on the results of morphological characteristics and electromagnetic parameters of steel slag, as well as the initial self-healing temperature of basic bitumen, the feasibility of using steel slag as microwave absorbers in asphalt mixtures for self-healing purposes has been further understood. In addition, the optimal substitution content of steel slag can be confirmed based on various road performance tests. Accordingly, the following conclusions were drawn.

(1) The porous structure of steel slag and its elements component observed from SEM images and SEM–EDS results, showed the potential microwave absorbing properties of steel slag to transfer heat and heal cracks under the microwave irradiation. Besides, the electromagnetic parameters of steel slag also supported the idea that as a typical magnetic material, steel slag can convert microwave energy into more thermal energy.

(2) The initial self-healing temperature (46.4 °C) of basic bitumen was acquired based on the results of frequency sweep analysis via DSR, and it was slightly lower than the softening point of 47.8 °C. At this threshold, the asphalt binder can start to flow and heal cracks even when the ambient temperature was slightly lower than its softening point.

(3) In order to confirm the optimal substitution of steel slag, the road performance of steel slag based asphalt mixtures was estimated. Results showed that the addition of steel slag in asphalt mixtures can help to increase the high-temperature stability, however, it had negative effects on both low-temperature crack resistance and water stability properties mainly due its porous structure. Thus, there exists an optimal steel slag content to obtain better road performance, and in this study, a substitution of 60% limestone aggregate with steel slag under the particle size of 4.75–9.5 mm by volume was suggested.

## Figures and Tables

**Figure 1 materials-13-00663-f001:**
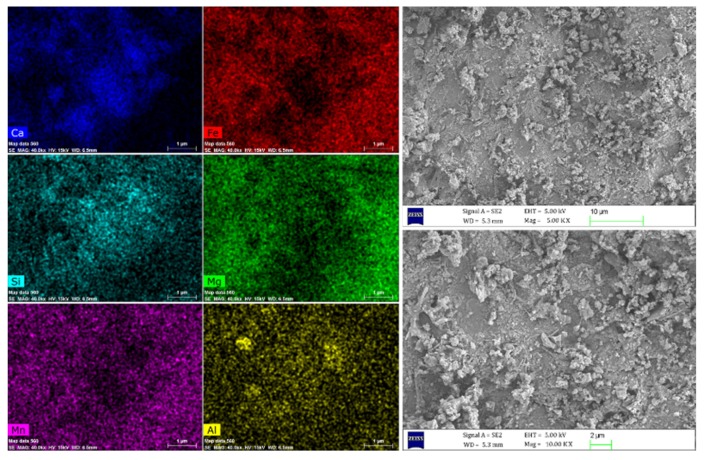
Map-scanning of steel slag surface and SEM results.

**Figure 2 materials-13-00663-f002:**
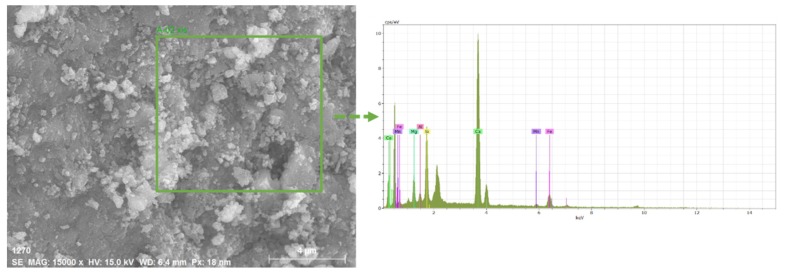
SEM–electron dispersive X-ray spectroscopy (EDS) analysis results of the steel slag specimen.

**Figure 3 materials-13-00663-f003:**
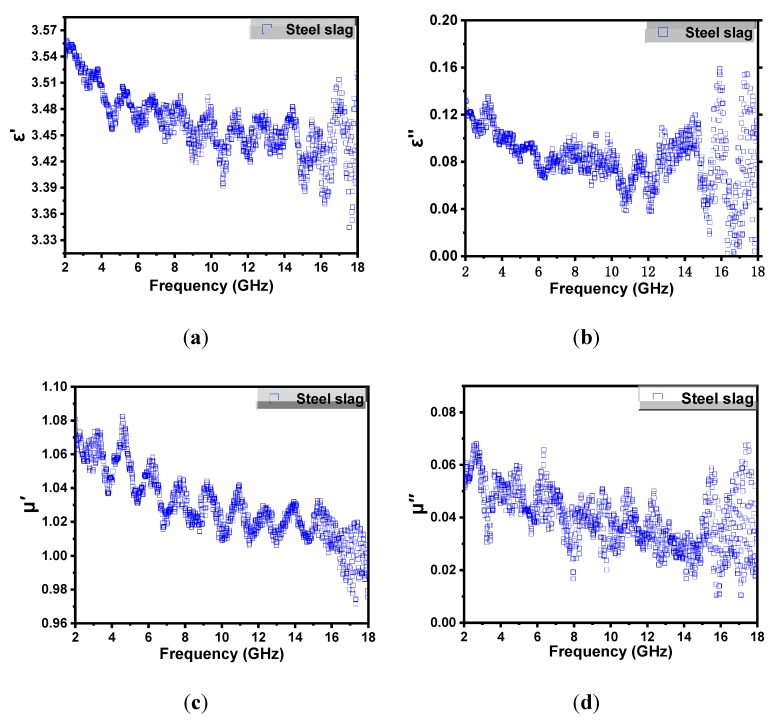
Electromagnetic parameters of steel slag. The real (**a**) and imaginary (**b**) parts of permittivity; the real (**c**) and imaginary (**d**) parts of permeability.

**Figure 4 materials-13-00663-f004:**
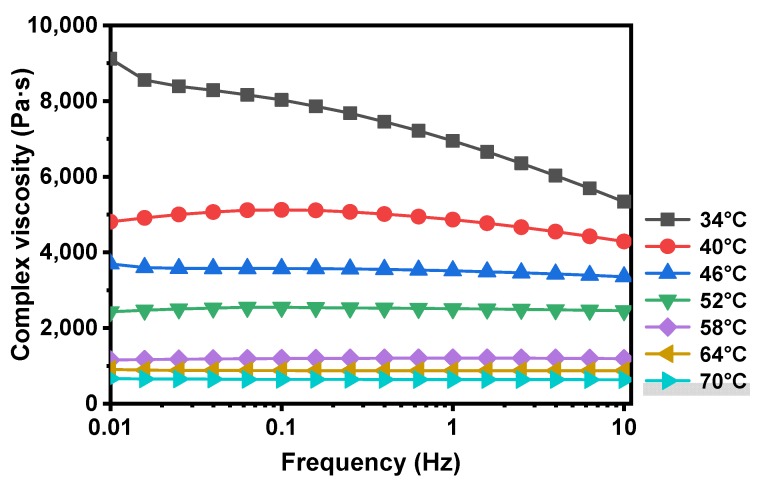
Frequency-complex viscosity relationship of the basic bitumen under different temperatures.

**Figure 5 materials-13-00663-f005:**
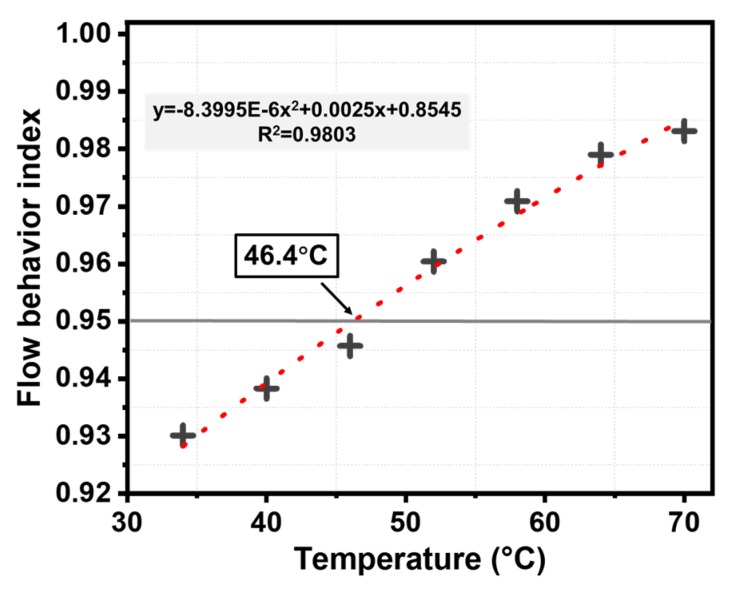
Flow behavior index of the basic bitumen.

**Figure 6 materials-13-00663-f006:**
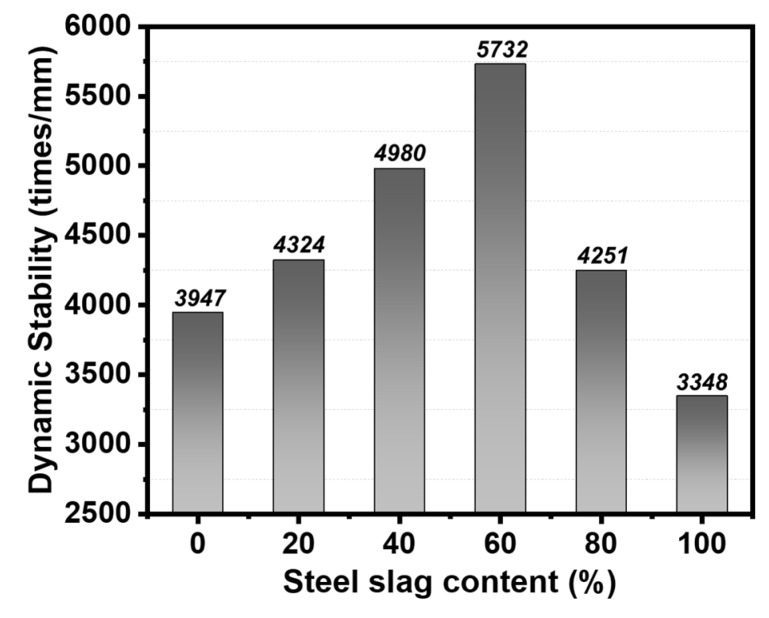
High temperature performance of different steel slag contents.

**Figure 7 materials-13-00663-f007:**
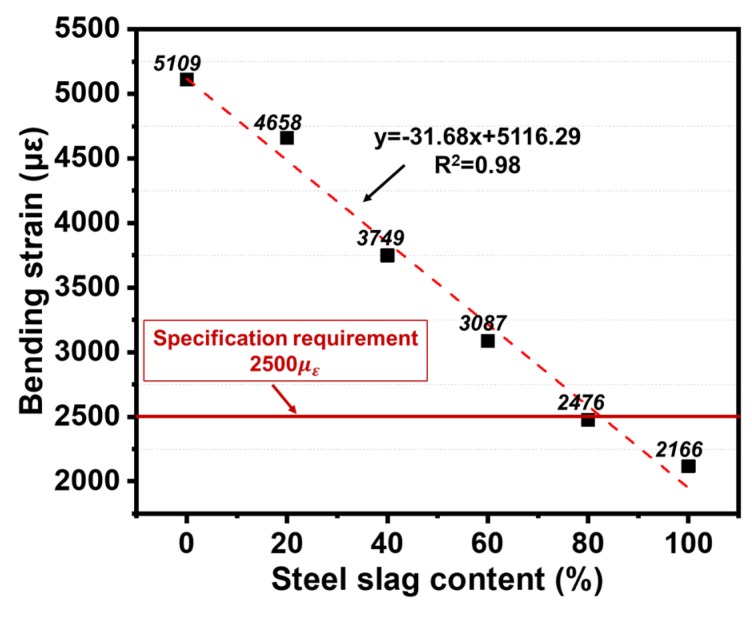
Low temperature performance of different steel slag contents.

**Figure 8 materials-13-00663-f008:**
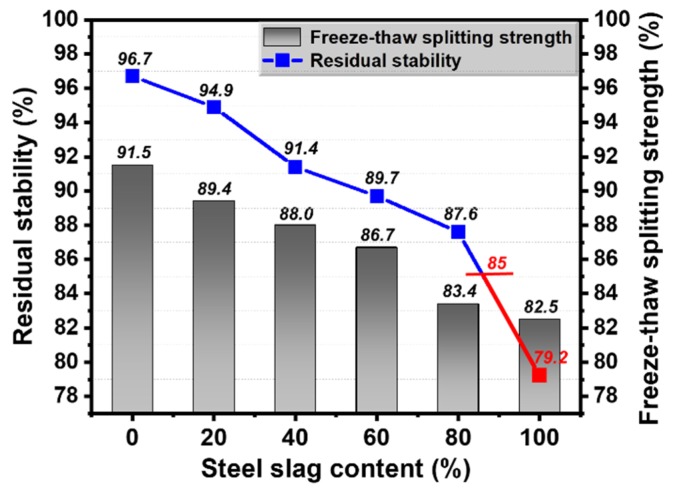
Water stability of different steel slag contents.

**Table 1 materials-13-00663-t001:** Particle size distribution of aggregates.

Sieve Size (mm)	Passing Rate (%)
13.2	95
9.5	76
4.75	53
2.36	37
1.18	27
0.6	19
0.3	13
0.15	10
0.075	5

**Table 2 materials-13-00663-t002:** Basic characteristics of aggregates used.

Property	Size (mm)	Relative Bulk Specific Gravity (g/mm^3^)	Los Angeles Abrasion Value (%)	Water Absorption (%)
Limestone	4.75–9.5	2.782	24.8	0.61
Steel slag		3.67	15.4	1.01

**Table 3 materials-13-00663-t003:** Mix proportion for Marshall specimens.

Mixture Type	Bitumen (by Mass)	% of Addition under 4.75–9.5 mm Particle Size (by Volume)	
		Limestone	Steel Slag
1	5.0%	100	0
2		80	20
3		60	40
4		40	60
5		20	80
6		0	100

**Table 4 materials-13-00663-t004:** Mass concentration of main elements in steel slag.

Element	Ca	Fe	Si	Mg	Mn	Al
Concentration (%)	62.83	17.19	11.71	4.66	2.49	1.12

**Table 5 materials-13-00663-t005:** Fitted results of flow behavior index at different temperatures.

Temperature (°C)	Fitting Formula	n − 1	Flow Behavior Index
34	y = 6757x^−0.0699^	−0.0699	0.9301
40	y = 4761x^−0.0617^	−0.0617	0.9383
46	y = 3486x^−0.0543^	−0.0543	0.9457
52	y = 2502x^−0.0396^	−0.0396	0.9604
58	y = 1197x^−0.0291^	−0.0291	0.9709
64	y = 875x^−0.0210^	−0.0210	0.9790
70	y = 640x^−0.0169^	−0.0169	0.9831

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
