# Peer review of "Microwave Absorption Ability of Steel Slag and Road Performance of Asphalt Mixtures Incorporating Steel Slag"

_materials, 2020, doi:10.3390/ma13030663_

Round 1

Reviewer 1 Report

This is an interesting study on the effects of steel slag being used as an additive to the aggregates in asphalt concrete. Following are the comments which may be incorporated for further improvement:

Minor revision with a grammatical check and spell check is required. Do we have any actual project built using this technique? If yes then can you include some details of that project. At some points, you have used "bitumen" and at some points, you have used "asphalt". Please elucidate on this. The conclusion section needs to be re-written/re-phrased.

Reviewer 2 Report

Tests evaluating the ability of slag to seal road cracks should be incorporated. This capacity is evaluated at the aggregate (slag) level, but not when the mix has already been executed and cracked. The mixtures with the different percentages of slag should be tested to check the validity of the slag in this type of maintenance treatment. The above comment leads to the fact that there is no clear definition of the element being tested: slag or asphalt mix with slag. If it is the asphalt mix that is being tested, the tests carried out on the slag (SEM, microwave absorption and initial self-healing temperature) should be introduced in the section related to the characterisation of the materials. With crack sealing tests on the developed mixtures, these tests performed on the slag and bitumen would make sense. As these tests have not been carried out, the tests carried out on the aggregate are not linked to the tests on the mixtures, the article being divided into two unrelated parts.

In relation to the manuscript, some necessary changes are:

Table 1 is completely unnecessary. To enter the properties of a single material, you must enter them in the text. The granulometry is presented in table 2 that is complex to interpret easily. A graph is much more suitable for representing the grain size of the material. The dosage of the mixtures carried out does not appear. The percentage of bitumen used and the percentage of slag are indicated, but not the fractions of natural aggregates used. The dosage of the mixtures must always be indicated in a table where the quantities of each material used are given for each mixture. The section regarding the physical properties of the materials should be improved. There is no reference to the test standards. The time period considered for water absorption is not indicated and the values for the standard time of 24 hours seem low. There is no indication of the freeze-thawing process to which the samples of the mixtures were subjected. It is only indicated that the test samples were subjected to it. Figure 2 needs to be improved in quality and size so that the graph can be seen clearly. Table 5 contains information that is not relevant. The adjustment formula used has already been discussed in the text above, so it is not necessary to indicate the formula for each temperature. Furthermore, the value "n-1" has no physical or engineering meaning, it is only a means of calculating the Flow behavior index. It is sufficient to indicate the Flow behavior index for each temperature in figure 4. The references should be adapted to the style required in the journal and be uniform. The most important error is that the name of some journals is indicated in an abbreviated form, while in others the full name is indicated without an abbreviation.

Reviewer 3 Report

The reviewer would like to thank the authors for their efforts. Paper has good organization and flow.

The Introduction provides sufficient background and in reviewer opinion the problem is clearly presented.

The Materials and Methods are sound and fulfill the requirements. 

Results and discussion paragraph clearly presented properties of asphalt mixtures with use of steel slag material. The results showed the great potential arising from the use of steel slag.

Reviewer 4 Report

"Introduction"

In civil engineering "distress" should be used instead of the "desease".

Expression as "a large number experiments..." or "many researchers gave made great efforts..." should be based on the large number of citations of the newest articles. Now one sentence is based on 1 or 2 cited authors.

Introduction must be improved with analysis of the newest researches in this topic (in the introduction should be given what is done and chronology of development in this topic, what is missing and what should be done further). The novelty of the experimental research should be based on this review. 

"Test methods and materials"

Standartized test methods as wheel tracking test or water stability could be decribed in summarized style (table or similar).

"Result"

How the optimal content of steel slag was determined?

"Conlusions"

In the conclusion should be revealed what was found from the experimental reseach, what is the most important.

Round 2

Reviewer 2 Report

A single point: a curve allows to see if the granulometry of the aggregate is continuous or discontinuous and its potential adjustment to reference curves (Bolomey, Fuller...). A table does not offer this possibility. 

I encourage you to address the aspects dealt with in the slag regarding crack sealing crack in the mixtures themselves. It would be, from my point of view, a very new and interesting aspect

Reviewer 4 Report

Thank you for the considering reviewer's comments.

Few photos of your used materials or testing procedure would be attractive for the readers.